# Beneficial Effects of Mindfulness-Based Stress Reduction Training on the Well-Being of a Female Sample during the First Total Lockdown Due to COVID-19 Pandemic in Italy

**DOI:** 10.3390/ijerph18115512

**Published:** 2021-05-21

**Authors:** Alessandra Accoto, Salvatore Gaetano Chiarella, Antonino Raffone, Antonella Montano, Adriano de Marco, Francesco Mainiero, Roberta Rubbino, Alessandro Valzania, David Conversi

**Affiliations:** 1Department of Psychology, Sapienza University of Rome, 00185 Rome, Italy; accotoa@gmail.com (A.A.); salvatoregaetano.chiarella@uniroma1.it (S.G.C.); antonino.raffone@uniroma1.it (A.R.); demarco.1598754@studenti.uniroma1.it (A.d.M.); mainiero.1690642@studenti.uniroma1.it (F.M.); 2School of Buddhist Studies, Philosophy and Comparative Religions, Nalanda University, Rajgir 803116, India; 3A.T. Beck Institute, 00185 Rome, Italy; antonellamontano@istitutobeck.it (A.M.); robertarubbino86@gmail.com (R.R.); alessandro.valzania@gmail.com (A.V.)

**Keywords:** MBSR, mindfulness meditation, COVID-19, women’s mental health, well-being, self-reports, psychological flexibility, internet-based intervention, mediation analysis

## Abstract

The global pandemic caused by COVID-19 and the subsequent lockdown have been widely recognized as traumatic events that pose threats to psychological well-being. Recent studies reported that during such traumatic events, women tend to be at greater risk than men for developing symptoms of stress, anxiety, and depression. Several studies reported that a mindfulness-based stress reduction protocol (MBSR) provides useful skills for dealing with traumatic events. In our study, a sample of Italian females received an 8-week MBSR course plus 6 weeks of video support for meditation practice during the first total lockdown in Italy. We assessed the participants with questionnaires before and after this period to investigate their mindfulness skills, psychological well-being, post-traumatic growth, and psychological flexibility. After the intervention, the meditators group reported improvement in measures associated with self-acceptance, purpose in life, and relation to others compared to the control group. Furthermore, our results showed that participants with greater mindfulness scores showed high levels of psychological flexibility, which in turn was positively associated with higher levels of psychological well-being. We concluded that the MBSR could support psychological well-being, at least in female subjects, even during an unpredictable adverse event, such as the COVID-19 lockdown, by reinforcing key psychological aspects.

## 1. Introduction

At the end of December 2019, a new virus belonging to the Coronaviridae family called SARS-CoV-2 infected thousands of people in Wuhan in the Hubei province of China. This new coronavirus causes a contagious respiratory syndrome named COVID-19 [1,2]. From Wuhan, the virus rapidly spread all over China, and a few weeks later, many countries were affected by the COVID-19 epidemic. The quick spreading of the virus induced many governments to declare a state of emergency in which citizens were subjected to quarantine. On 30 January 2020, it was declared a public health emergency of international interest [3]. Since February 2020, the Italian Government, after considering the first COVID-19 case identified in the country [4,5], implemented hygiene guidelines and social practices to prevent the spread of the virus. Such national rules, issued through decrees of the President of the Council of Ministers (DPCM), were increasingly stringent, reaching a peak on 9 March 2020, when the Italian Government announced a strict national lockdown. Two distinct phases characterized the national lockdown in Italy. A complete total lockdown characterized the first phase (i.e., phase 1, from 9 March to 3 May 2020), with 56 million people estimated to be forced to stay at home; the second phase (i.e., phase 2, from 4 May to 24 June 2020) was characterized by loosening restrictions such that citizens could visit relatives or do exercise outside. The forced quarantine due to the lockdown brought massive changes in people’s daily lives, and the first studies showed the start of a parallel epidemic of depression, anxiety, and stress symptoms [6,7,8]. As many nations subsequently used the same strategy, many researchers worldwide started to investigate the psychological consequences of quarantine and social distancing [9,10,11]. Early studies showed that the adult population experienced an increase in psychopathological symptoms and a reduction in psychological well-being [12,13]. Studies on the psychological impact of COVID-19 on the Italian population found adverse effects, such as clinically significant post-traumatic symptoms and high psychological distress [14,15,16]. Recent studies underlined that women had worse mental health consequences and a greater psychological impact due to COVID-19 [17,18,19]. All these findings are in line with previous studies that proposed restricted movements, trauma exposure, perceived personal or family risk, and limited resources and information as underlying these negative psychological effects [20]. Moreover, the risks of infecting others or being infected imply unpredictability and uncontrollability [21,22], which are core features of the psychobiological construct of stress [23,24]. Other studies demonstrated an increase in maladaptive thoughts about the consequences of the pandemic [25,26,27], which, in turn, were associated with a poor ability to stay focused in the present moment [28]. 

Considering that the COVID-19 pandemic is characterized by uncertainty, psychological distress, and lack of visibility of the future, its harmful impacts should be promptly faced with systematic psychological self-care [29,30,31,32,33]. Mind–body meditation techniques, such as mindfulness meditation practices, may represent an adequate protective factor in such a critical situation. Several studies showed that these practices alleviate the stress, anxiety, and depression symptoms, inducing subjective changes in meditators that result in improved cognitive functioning and general well-being [34,35,36,37]. Mindfulness is defined as the individual tendency to focus attention and awareness on the current experience [38]. Instead, mindfulness meditation is a technique described as a particular way to paying attention in an open and non-judgmental way, focusing on bodily sensations and mental events that occur in the present moment, with the aim of cultivating equanimity, awareness, compassion, and self-acceptance [34,39,40]. Several studies in the last three decades showed that this practice leads to many psychological benefits in meditators, such as better self-control [41], enhanced flexibility [42,43], equanimity [44], improved concentration and mental clarity [45,46], emotional intelligence, and the ability to relate to others and oneself with kindness [47,48,49], self-acceptance, and compassion [50,51]. 

Mindfulness meditation is based on meditation techniques derived from the Buddhist tradition, such as “Vipassana” or insight meditation from the Theravada tradition. this practice is based on the mindfulness-based stress reduction protocol (MBSR) [52], which has become the most used and empirically supported mindfulness-based intervention [53,54]. Although some studies have recently reported adverse effects of the mindfulness-based intervention [55], many studies stated that the MBSR improves psychological health and reduces negative symptoms [35,56,57]. For instance, Carmody and Baer found a correlation between mindfulness skills improvement and psychological symptoms reduction [58]. Crucially, the MBSR protocol appears to be effective in handling stressful events [59,60], e.g., it helps people to manage post-traumatic stress disorder (PTSD) symptoms [61] or cope with cancer [62,63,64]. 

Recent studies reported psychological benefits, even when the MBSR was delivered through meditation apps or online classes [65,66,67] and, more recently, the effectiveness of MBSR courses in managing psychological suffering when face-to-face meetings are suspended were reported during the COVID-19 pandemic [68,69,70]. However, it is not clear which components of psychological well-being have been influenced by the effects of the pandemic and, therefore, can benefit more from the MBSR intervention. For instance, previous studies showed that, when circumstances seem too overwhelming and unconquerable, people are at risk for feeling unable to pursue their goals or contribute to others, especially if they lack emotional and informational support from others [71]. In these situations, high levels of self-acceptance can make traumatized people feel highly positive in the face of difficulties and setbacks [72], which may lead them to focus more on positive changes after trauma [73]; this is a phenomenon reported by Tedeschi and Calhoun and is known as post-traumatic growth [74]. It would therefore be interesting to understand what the COVID-19 pandemic has damaged and, in contrast, what its positive implications have been. A further question concerns which variables, following the MBSR intervention, are directly responsible for the beneficial effect on psychological well-being [67]. It has been suggested that, during stressful situations, the increase in MBSR-induced flexibility could represent an important variable underlying the relation between mindfulness skills and improved mental health [75]. The concept of flexibility includes several transdiagnostic personal skills. It is defined as being aware of the present moment and accepting the emotions, sensations, and thoughts [76,77]. It can also be considered central to human adaptation, well-being, and life satisfaction [76,77]. It was also demonstrated that flexibility ameliorates the negative consequences of stressors on mental health [76,77]. 

In the present study, an 8-week MBSR program was delivered during the outbreak of the COVID-19 pandemic in Italy. The first six meetings took place in groups (before the total lockdown), while the last two took place via the internet, and involved a sample of women. After the end of the 8-week MBSR program, participants received meditation video lessons for six further weeks. We recruited a control group in which no meditation program was delivered. Although the MBSR program was not originally aimed at mitigating the psychological consequences of the COVID-19 pandemic, as this event was unpredictable, it was subsequently investigated to test a highly vulnerable class of the general population. Indeed, as mentioned above, many studies have shown that being a woman is one of the risk factors for poor mental health outcomes during the COVID-19 pandemic [15,77]. 

With the aim of evaluating the effects of an MBSR program during the outbreak of an unpredictable negative event, such as the lockdown caused by the COVID-19 pandemic, four objectives were pursued in our study: to evaluate whether it is a useful tool (a) to develop mindfulness skills and (b) to overcome the psychological distress, (c) to evaluate positive reactions to negative events, and (d) to explore the psychological mechanism underlying the positive relationship between the MBSR and psychological well-being during the lockdown. 

## 2. Materials and Methods

### 2.1. Participants

Twenty-six participants took part at this study (mean age = 39.9 years; SD = 13.3 years). All participants were Italian women with no previous experience of meditation. All participants reported their socio-demographic (gender, age, nationality), health (whether they were under psychological or pharmacological treatment), and meditation (whether they practiced meditation) information. Each participant reported no clinical diagnosis or history of either psychiatric, neurological, or neurodevelopmental disorders and no use of drugs or substances acting in the central nervous system. A total of 15 (mean age = 41.7 years, SD = 15.2 years) participants took part in the MBSR group and 11 participants (mean age = 38.2 years, SD = 11.5 years) formed the control group. The participants self-selected their participation in the MBSR group or the control group, i.e., all the participants, including those in the control group, initially appeared at Rome’s Beck Institute to generically request a group treatment to improve stress management. Self-selection was limited to the choice between the MBSR or yoga. The yoga group should have taken place after the MBSR group, but it never started for reasons related to the pandemic’s evolution. At this point, the subjects who should have done yoga were used as the control group, equivalent to a waiting list. Informed written consent was obtained from all participants. 

### 2.2. Instruments 

To evaluate the MBSR’s effects, we administered the following four questionnaires.

Mindfulness skills were assessed using the Italian version of the Five Facet Mindfulness Questionnaire (FFMQ) [54,78]. This 39-item instrument measures five mindfulness scales: observing (OBS), describing (DES), acting with awareness (AWA), non-judging of inner experience (NJU), and non-reactivity to inner experience (NRE). Respondents are asked to rate each statement using a 5-level Likert scale (1—never or very rarely true, 5—very often or always true). Higher scores in each scale reflect a greater level of mindfulness. In the present study, the FFMQ scales showed Cronbach’s alpha values ranging from 0.60 to 0.74.

Psychological well-being was assessed with the Italian version of the 84-item Psychological Well-Being Scales (PWB) [79,80]. This tool uses a 6-level Likert scale response format (1—strongly disagree, 6—strongly agree). The six scales included are self-acceptance (SA), autonomy (AU), environmental mastery (EM), personal growth (PG), positive relations (PR), and purpose in life (PL). Higher scores on any scale indicate greater indices of happiness and psychological well-being. In the present study, the PWB showed Cronbach’s alpha values ranging from 0.60 to 0.80.

Psychological growth following exposure to traumatic events (we asked the participants to think of COVID-19 as a traumatic event) was assessed with the Italian version of the 21-item Post-Traumatic Growth Inventory (PTGI) [74]. The PTGI is scored using a 6-point Likert scale (1—no changes, 6—very large changes). The five scales included are new possibilities (NP), relation to others (RO), appreciation of life (AL), personal strength (PS), and spiritual changes (SC). Higher scores indicate greater growth. In the present study, the PTGI scales showed Cronbach’s alpha values ranging from 0.65 to 0.79. This questionnaire was administered only at T1 because it asked participants to think about their experience with COVID-19 lockdown. It, therefore, contained a measure of the change that had taken place from the pre-COVID-19 condition to the post-COVID-19 condition.

To investigate whether variables such as flexibility may play a role in determining the effect on well-being induced by the MBSR training, psychological flexibility was assessed by administering the Italian version of the 10-item self-report Acceptance and Action Questionnaire-II (AAQ-II) [80,81] to the MBSR group. The AAQ-II is scored using a 7-point Likert scale (1—never true, 7—always true), and higher scores indicate higher global psychological inflexibility. The Italian version of the AAQ-II [82] is a reliable and valid measure of psychological inflexibility, with high internal consistency (0.83). In the present study, Cronbach’s alpha was 0.81.

### 2.3. Procedure

The Beck Institute, which is a primary source for therapy, training, and resources in cognitive behavior therapy (CBT), staged an 8-week MBSR training program. The MBSR was led by the Beck Institute’s manager (A.M.), one of the authors of this paper. She is an experienced psychologist and mindfulness meditation practitioner. This project was originally developed by the Sapienza University of Rome and was designed to study the differences between the MBSR and control participants in stress-related microRNAs through the collection of two salivary samples at two different times, T0 (pre-MBSR: between 20 and 28 January) and T1 (post-MBSR: between 27 April and 4 May). However, due to the pandemic’s outbreak in the sixth week of the MBSR course, it was not possible to collect saliva samples after the training. 

The MBSR training program started on 29 January 2020. Before starting the course (T0), both the MBSR and control groups were administered the FFMQ and PWB, whereas only the MBSR group was administered the AAQ-II. At the end of the sixth week of the MBSR course, a lockdown all over Italy was in place during which citizens were only allowed to go outside for a justified reason and with only one other person or the core family. On 11 March 2020, the Italian government ordered the closure of all universities and schools, leisure places, and public buildings all over Italy. Employees were asked to work in their home office and therefore live activities of the Beck Institute and Sapienza University were unable to occur. Therefore, the remaining two MBSR lessons (week 7 and 8) were delivered through online video lessons via the internet. During the 8-week MBSR, the home practice was assessed through diaries collected every two weeks. After the end of the MBSR program, the Beck Institute decided to support participants through online video meditation lessons. Those lessons lasted until the end of the lockdown, i.e., for another six weeks. At this time (T1), we administered the FFMQ, PWB, and PTGI to both groups, whereas only the MBSR group was administered the AAQ-II. The timeline of the events is represented in Figure 1.

During these additional six weeks, participants could communicate with the project managers and their instructors via e-mail or telephone, share their meditation experiences, or ask questions about the video lessons’ contents. Participants were assessed in the days before the MBSR training program (T0) through paper and pencil questionnaires and at the end of the lockdown (T1) using internet-based questionnaires. The choice of this timing was motivated by the need to test the effects of the MBSR training before the partial reopening occurred in the next phase of lockdown (i.e., phase 2 of lockdown). The Sapienza University Ethics Committee approved the study protocol, and the procedure was undertaken according to the ethical standards laid down in the 1964 Declaration of Helsinki. All participants showed almost total adherence to all MBSR meetings.

### 2.4. Statistical Analyses

A preliminary analysis of the between-group differences at baseline was analyzed via the FFMQ total score and the PWB total score using independent-sample *t*-tests. To investigate the effects of the MBSR on the FFMQ and PWB outcomes, we implemented a 2 × 2 repeated-measures ANOVA with time (2 levels: T0 vs. T1) as the within-subjects factor and group (2 levels: MBSR vs. control) as a between-subjects factor. As a measure of effect size, the partial eta-squared (η_P_^2^) was used for the main effects in ANOVA analyses and Cohen’s d was used for the post-hoc and *t*-test analyses. All post-hoc *t*-tests were corrected with the Bonferroni method. Between-group comparisons on PTGI scales were conducted using independent-sample *t*-tests for each subscale. Within-group comparisons on the AAQ-II scale were conducted using paired-sample *t*-tests. Pre-to-post changes in the psychological flexibility in the MBSR group were tested using a paired-sample *t*-test. A mediation analysis tests the effect of a variable (flexibility) that accounts for the relation between a predictor variable (mindfulness skills) and an outcome variable (psychological well-being) [83]. All mediation analyses were conducted using the PROCESS macro in SPSS (Statistical Package for Social Science, version 26). Bootstrapping (1000 resamples) with bias-corrected and accelerated 95% confidence intervals (BCa) were used to adjust for measurement error. For this analysis, the total score on the FFMQ questionnaire was considered as a predictor and the global psychological well-being index of the PWB questionnaire was taken as a criterion variable. The mediated relationship of mindfulness and mental health via flexibility was then calculated. A mediator mediates the relationship between the independent and dependent variables explaining the reason for such a relationship to exist. 

## 3. Results

### 3.1. Preliminary Analyses: The Between-Group Difference at Baseline

Between-group differences at baseline were analyzed using independent-sample *t*-tests to test for the initial equivalence of the groups. The independent-sample *t*-test on the FFMQ total score and its subscales revealed no difference between the MBSR and control groups at baseline, except for the subscale NRE in which the controls’ score was higher. 

The independent-sample *t*-test with the PWB total score and its subscales revealed a difference between the MBSR and control groups at baseline, with the higher score for the control group compared to the MBSR group for the following scales: total score; PG, PR, PL, and SA. All data and *t*-test analyses are reported in Appendix A. 

### 3.2. Effects of MBSR on Mindfulness Variables Measured Using the FFMQ Questionnaire

A first 2 × 2 repeated-measure ANOVA on the FFMQ total score revealed that the time × group interaction effect reached significance (F(1,24) = 5.08, *p* = 0.034, η_P_^2^ = 0.17). Post-hoc tests with Bonferroni corrections revealed a difference between T0 and T1 in the MBSR group (t(25) = 3.20, *p* = 0.023, d = 0.63), with a higher score at T1 compared to T0, while there was no difference between T0 and T1 in the control group (t(25) = −0.22, *p* = 1, d = −0.04) (see Figure 2a). The results also showed that there was no difference between the MBSR and control groups at time T0 (t(25) = −1.22, *p* = 1, d = −0.24), thus confirming no difference between the groups at baseline.

Then, we ran the same ANOVA on all the subscales of the FFMQ. The results revealed a significant main effect of time on the NRE subscale, (F(1,24) = 20.24, *p* < 0.001, η_P_^2^ = 0.46), with higher scores at T1 compared to T0. The time × group interaction effect reached significance (F(1,24) = 10.10, *p* = 0.004, η_P_^2^ = 0.30). Post-hoc tests with Bonferroni corrections revealed a significant difference between T0 and T1 in the MBSR group (t(25) = 5.90, *p* < 0.001, d = 1.16), with a higher score at T1 compared to T0, with no difference between T1 and T0 in the control group (t(25) = 0.87, *p* = 1, d = 0.17) (see Figure 2b). Results also showed that there was no difference between the MBSR and control groups at time T0 (t(25) = −2.58, *p* = 0.08, d = −0.23). No differences were found in the other subscales.

### 3.3. Effects of MBSR on Psychological Well-Being Variables as Measured Using the PWB Questionnaire

Like for the FFMQ, we ran the same 2 × 2 repeated-measure ANOVA on the PWB total score, which revealed a significant main effect of time (F(1,24) = 10.91, *p* = 0.003, η_P_^2^ = 0.31) with an increase in the scores from T0 to T1. The main effect of group was not significant (F(1,24) = 3.18, *p* = 0.090, η_P_^2^ = 0.11). Furthermore, the time × group interaction effect reached significance (F(1,24) = 43.24, *p* < 0.001, η_P_^2^ = 0.64). Post-hoc tests with Bonferroni corrections revealed a difference at T0 between the MBSR and control groups (t(25) = −3.20, *p* = 0.021, d = −0.63), with higher scores in the control group compared to the MBSR group and a difference between T0 and T1 in the MBSR group (t(25) = 7.59, *p* < 0.001, d = 1.49), with higher scores in the PWB measures in T1 compared to T0. No differences were found between T0 and T1 in the control group (t(25) = −2.15, *p* = 0.249, d = −0.42) (see Figure 3a). 

The same ANOVA on the PWB subscales revealed a main effect of time (F(1,24) = 6.86, *p* = 0.015, η_P_^2^ = 0.22) in the PG subscale, showing an increase in the scores from T0 to T1. Furthermore, the main effect of group was significant (F(1,24) = 19.90, *p* < 0.001, η_P_^2^ = 0.45), with higher scores in the control group compared to the MBSR group. The analysis also revealed that the time × group interaction effect reached significance (F(1,24) = 10.94, *p* = 0.003, η_P_^2^ = 0.31). Post-hoc tests with Bonferroni corrections revealed differences between the MBSR and control groups at T0 (t(25) = −5.48, *p* < 0.001, d = −1.07), with higher scores in the control group compared to the MBSR group and between T0 and T1 in the MBSR group (t(25) = 4.55, *p* < 0.001, d = 0.89), with a higher score at T1 compared to T0. The analysis did not reveal any differences between T0 and T1 in the control group (t(25) = −0.45, *p* = 1.00, d = −0.09) (see Figure 3b). 

The ANOVA on the PR subscale revealed a significant main effect of time (F(1,24) = 4.87, *p* = 0.037, η_P_^2^ = 0.17), with an increase in the scores from T0 to T1. Moreover, the main effect of group was significant (F(1,24) = 19.07, *p* < 0.001, η_P_^2^ = 0.44), with higher scores in the control group compared to the MBSR group. The time × group interaction effect reached significance (F(1,24) = 7.55, *p* = 0.011, η_P_^2^ = 0.24). Post-hoc tests with Bonferroni corrections revealed differences between the MBSR and control groups at T0 (t(25) = −5.11, *p* < 0.001, d = −1.00), with higher scores in the control group compared to the MBSR group, and between T0 and T1 in the MBSR group (t(25) = 3.80, *p* = 0.005, d = 0.74), with higher scores in the PWB measures at T1 compared to T0. The analysis did not reveal any differences between T0 and T1 in the control group (t(25) = −0.36, *p* = 1.00, d = −0.07) (see Figure 3c). 

The ANOVA on PL revealed that the time × group interaction effect reached significance (F(1,24) = 31.20, *p* < 0.001, η_P_^2^ = 0.56). Post-hoc tests with Bonferroni corrections revealed differences between T0 and T1 (t(25) = 5.11, *p* < 0.001, d = 1.00) in the MBSR group, with higher scores in the PWB measures at T1 compared to T0, and between T0 and T1 in the control group (t(25) = −2.98, *p* = 0.039, d = −0.58), with lower scores in the PWB measures in T1 compared to T0 (see Figure 3d).

The ANOVA on the SA subscales revealed that the time × group interaction effect reached significance (F(1,24) = 30.04, *p* < 0.001, η_P_^2^ = 0.56). Post-hoc tests with Bonferroni corrections revealed differences between T0 and T1 in the MBSR group (t(25) = 4.95, *p* < 0.001, d = 0.97), with higher scores in the PWB measures at T1 compared to T0, and between T0 and T1 in control group (t(25) = −2.97, *p* = 0.04, d = −0.58), with lower scores in the PWB measures at T1 compared to T0 (see Figure 3e).

All data and analyses are reported in Appendix A.

### 3.4. Effects of MBSR on Post-Traumatic Growth Variables Measured Using the PTGI Questionnaire

Independent *t*-tests showed significant differences between the MBSR and control groups in the PTGI RO subscale. The MBSR group reported a higher score in the RO subscale compared to the control group (t(24) = 2.07, *p* = 0.049, d = 0.82) (see Figure 4). All data and analyses are reported in Appendix A.

### 3.5. Effects of MBSR on Flexibility as Measured Using the AAQ-II Questionnaire

Paired-sample *t*-tests showed significant differences between T0 and T1 in the AAQ-II total score in the MBSR group. The scores obtained after the MBSR training were higher than those detected before the training (t(24) = 2.20, *p* = 0.045, d = 0.56) (see Figure 5). All data and analyses are reported in Appendix A.

### 3.6. Mindfulness Predictor of Psychopathological Outcomes during the Pandemic in MBSR Group 

The procedure proposed by Barron and Kenney was first employed to test for the mediation of flexibility on the relation between mindfulness and psychological well-being after the MBSR training [84]. First, there was a significant association between the predictor (mindfulness post-MBSR training as measured using the FFMQ questionnaire) and the outcome variable (psychological well-being post-MBSR training as measured using the PWB questionnaire) (β  = 0.459, *p* = 0.006). Such a relation was significant between the same variables measured before the MBSR training (β = 0.319, *p* = 0.028). Next, after controlling for the relationship between the predictor and the outcome, the proposed mediator (flexibility post-MBSR as measured using the AAQ-II questionnaire) was related to the outcome (β = −0.5911, *p* < 0.006). Such a relation did not exist between the same variables measured before the MBSR training (β  = 0.3720, *p* = 0.12). Finally, regarding the relation between mindfulness and psychological well-being after the MBSR training, it was no longer significant, which is indicative of total mediation (indirect mediation β = 0.99, *p* = 0.047; direct mediation β = 1.19, *p* = 0.063; total mediation β = 2.18, *p* = 0.005). Such mediation did not exist between the same variables measured before the MBSR training (indirect mediation β = −0.33, *p* = 0.29; direct mediation β = 1.95, *p* = 0.011; total mediation β = 1.62, *p* = 0.028). The results indicate that the effect of mindfulness on psychological well-being after the subjects participated in the MBSR training was mediated by the flexibility acquired because of the MBSR training, as this did not happen when the same analysis was conducted on the same variables measured before the MBSR training (see Figure 6). The model fit indices are reported in Appendix A. 

## 4. Discussion

In the present study, we investigated the effects of mindfulness-based training that took place during the first total lockdown due to the spread of the COVID-19 pandemic in Italy. We took advantage of having collected data before the COVID-19 outbreak using self-report questionnaires assessing mindfulness (FFMQ), psychological well-being (PWB), post-traumatic growth (PTGI), and flexibility (AAQ-II) in a sample of women. 

Participants attended an 8-week MBSR protocol, which was conducted partially online (i.e., the last two sessions). Moreover, they received 6 weeks of video assistance for mindfulness home practice until the end of the lockdown such that the period between the pre- and post-training was 14-weeks in total. The results of the MBSR participants were compared with those of a control group that did not receive any treatment during the same period. Although, initially, the purpose of the study was to investigate the effect of the MBSR on the modulation of specific molecular stress markers, the lockdown forced us to reshape the goal of the study. This unpredictable event allowed us to investigate the effectiveness of a reshaped MBSR program during the outbreak of a particularly negative event that was characterized by uncertainty and psychological distress [30,31,32,33,34]. 

### 4.1. Objective 1: Effects of MBSR Training on Mindfulness Skills

Although the poor ability to stay focused in the present moment due to being deeply affected by this period of extreme criticality could represent an obstacle to the MBSR’s efficacy [84,85], our results highlighted the effectiveness of the MBSR training in improving mindfulness skills, as demonstrated by the significant increase in the total score and non-react subscale of the FFMQ questionnaire. Previous research indicated positive effects of the MBSR program, including reduced stress and anxiety [86,87,88,89]. It was demonstrated that non-reactivity, defined as the capacity to choose not to react to emotions and negative thoughts and to accept their existence, is a protective factor against stress [90]. This ability will lead to a completely different downstream experience of the initial stimulus [90]. As such, it is possible that the non-reactivity facet plays a primary role in the process of mindfulness, above and beyond the remaining FFMQ facets [90,91]. This result represents the first step toward demonstrating the effectiveness of the MBSR training in facilitating the establishment of a mental setting based on mindfulness that could promote a functional approach to coping with stressful experiences, such as the COVID-19 pandemic. The absence of the detrimental impact of COVID-19 on the mindfulness measure in the control group, whose score remained unchanged from pre- to post-evaluation, could be explained by the lack of an impact of stressors on trait mindfulness that could represent a baseline condition that can counteract the stressful effects of the pandemic. In contrast, the COVID-19 emergency had a strong impact on psychological well-being, as shown by several studies. Data from previously published studies that investigated mental health during the COVID-19 pandemic clearly showed an increased mental health burden [92,93]. The lack of a daily routine and COVID-19-related fears and uncertainties may affect psychological well-being, including symptoms of depression, anxiety, sleep disorders, aggression, drug abuse, or even suicidal behavior [9]. 

### 4.2. Objective 2: Effects of MBSR Training on Psychological Well-Being

Therefore, we thought it might be useful to evaluate the efficacy of the MBSR training to support psychologically burdened people during the COVID-19 pandemic. The results also supported the effectiveness of the MBSR training; at the end of the program, immediately after phase 1 of the lockdown, significant improvements were observed in the MBSR group when compared to the control group in most of the self-reported measures of psychological well-being. The MBSR group had significantly improved in the PWB total score and the subscales of personal growth, positive relation, purpose in life, and self-acceptance. Moreover, our preliminary findings demonstrated not only a significant increase in many features of psychological well-being after the MBSR training but also a significant worsening in the purpose in life and self-acceptance subscales in the control group. Such results demonstrated the efficacy of the MBSR training in ameliorating well-being during the pandemic and the capability to overcome the burden of the pandemic on some psychological aspect. It is noteworthy in the context of research on the psychological consequences of COVID-19 that the measures that are most affected by the pandemic were those relating to the purpose in life and self-acceptance and that, despite this, they benefited from the MBSR treatment. In the specific context of the COVID-19 pandemic, Trzebiński and colleagues showed that a higher level of purpose in life (i.e., having a clear purpose and meaning in life, having life goals, not being afraid of the future) was related to lower anxiety and emotional distress during the crisis [94]. Therefore, the authors argued that purpose in life, among other factors, may work as a buffer against stress-related reactions to the pandemic [94]. Moreover, it was demonstrated that interpersonal issues, including domestic violence, abuse, trauma, negative emotions, unhealthy relations and family environment, economic problems, and poor health, in addition to the COVID-19 pandemic, exacerbated the well-being of all individuals, making them question their self-acceptance [95,96,97,98]. Therefore, self-acceptance becomes an important factor that buffers the negative effect of stressful life events, such as the COVID-19 pandemic. It can serve as a target for the prevention of negative health outcomes. 

### 4.3. Objective 3: Post-Traumatic Growth

In the third aim, we assessed the possibility of a positive reaction to negative events, such as the COVID-19 pandemic. We explored whether after experiencing an adverse situation, participants achieved personal growth due to the strengthening of resilient variables [99,100]. To make this happen, the event must question beliefs about oneself and others, as happened during the COVID-19 pandemic [74,101]. Tedeschi and Calhoun define post-traumatic growth as positive psychological changes (i.e., a greater sense of personal strength and closer relationships with others) that happen because of a person’s struggle with a traumatic event [74,101]. The authors propose that an individual reconstructs their beliefs and tries to promote constructive thinking through the regulation of their emotions to reconcile with the memories of the trauma [74,101]. Our results showed an improvement in the scores on the relation to others subscale in the MBSR group. This improvement was not present in the control group. This result did not allow us to attribute the improvement in the relation to others subscale to a positive reaction to the pandemic. Only meditators were likely able to benefit from relationships with others in the family environment thanks to the psychological skills provided by the meditation training. Nevertheless, the increase in the positive attribution to interpersonal relationships allowed us to further affirm how effective the MBSR training was in providing participants with adequate strategies to cope with stress.

Several studies reported that mindfulness meditation can modulate intersubjective and self–other representations [102,103,104] while promoting positive social behaviors, such as compassion, empathy, and altruism [105]. Moreover, the use of social strategies in mood modulation has previously been demonstrated by Ulusoy et al. [106]. The authors demonstrated the association between the use of instrumental social support and depressive disorders. Scores regarding social support were lower in patients with any depressive disorder than those without any depressive disorder [106]. Less use of these coping strategies may lead to perceptions of the continuity of the stressor or can increase the sense of helplessness [101]. The use of instrumental social support, such as relationships with others, can be an important coping strategy not only in depressive disorders but also in coping with anxiety and stress during high-criticality situations [106]. 

### 4.4. Objective 4: The Role of Psychological Flexibility as Mediator Variable 

Finally, we investigated the role of psychological flexibility in mediating the relationship between mindfulness skills (FFMQ) and psychological well-being (PWB). As predicted, global psychological flexibility mediated the impact of mindfulness skills on mental health when measured at T1. We did not observe such a mediation effect between the same variables measured at T0, prompting us to hypothesize that the MBSR training could induce the detected mediation effect acting on flexibility.

Due to the absence of a control group, we could not firmly conclude a causal effect of flexibility from our data. Nevertheless, our data are in line with the hypothesis that MBSR training improves psychological well-being by increasing psychological flexibility. The role of psychological flexibility as a protective psychological resource during a pandemic and the associated social restrictions is consistent with prior research showing that psychological flexibility is related to better mental health in a wide range of contexts and can be substantially improved using acceptance and commitment therapy (ACT) or mindfulness, or both [77,107,108,109,110,111]. Our results are consistent with prior studies that demonstrate a similar mediation effect of daily stress [107], learned helplessness, and major life events on mental health in the general population [112]. Nevertheless, our study is the first to our knowledge showing that the improvement in psychological flexibility during the MBSR training could mediate the effects of the intervention on positive mental health. Our study thereby provides the first evidence that mental health can be promoted by stimulating flexibility skills in a period of extreme criticality. 

### 4.5. Limitations

This study has significant limitations, partly relating to the unprecedented infectious pandemic it sought to exploit, which must be acknowledged and explained. First, online surveys and self-report measures are susceptible to socially desirable responses, potentially yielding biased results. This issue is particularly relevant because other studies found that MBSR participants self-reported decreased stress, whereas cortisol testing showed the opposite [113,114]. Therefore, future studies should supplement the self-report measures with evaluations of biological markers of the stress response, i.e., salivary cortisol. Second, the sample size was relatively small and composed only of females, limiting the findings’ generalizability. Third, the absence of randomization may have contributed to the differences in overall psychological measures observed at baseline between the MBSR and control groups. Moreover, the lack of a randomized control group and a propensity score analysis does not warrant a causal inference regarding the effect of the MBSR treatment. Indeed, we cannot exclude those changes attributed to the MBSR arose, at least partly, from a different initial propensity to treatment or generic components of group-based interventions, i.e., support, destigmatization, therapeutic attention, emotional expression in the groups, and other placebo effects. However, none of the above limitations are compelling a priori reasons to assume a different initial propensity since subjects from both groups initially sought assistance to improve stress management through mind–body techniques. A possible speculative explanation is that the control subjects, who initially agreed to practice yoga, had a greater propensity to use the body as a tool to achieve well-being. Such a propensity may be associated with better physical and mental health, as suggested by the higher scores in psychological measures detected in the control group at T0. Accordingly, participants who self-selected for the MBSR training reported psychological well-being levels at baseline that were lower than those of the controls. Fourth, the higher baseline levels of psychological well-being reported by the control subjects may have caused a ceiling effect, i.e., they may have prevented the observation of an improvement due to the passage of time. However, the control subjects reported a worsening in the PWB subscales of “self-acceptance” and “purpose in life,” ruling out a ceiling effect for these variables. Thus, the MBSR may have at least prevented such a worsening in participants. 

Although we found flexibility to be a statistically significant mediator of the MBSR’s effects on well-being, our mediation analysis lacked a comparison with a non-treated control group. Therefore, our results do not warrant any causal inference on the mediating role of flexibility, as previously stated. Future studies must measure the putative mediator in the treated and control groups to establish such a role. Nonetheless, our data are in line with previous evidence suggesting psychological flexibility as mechanisms underlying mindfulness-based interventions [56].

## 5. Conclusions

Our study provides preliminary evidence that mindfulness-based group training can support psychological well-being, at least in female subjects, even during an unpredictable adverse event, such as a pandemic-related lockdown. Moreover, our results suggest that the MBSR did so by targeting crucial aspects of the mindfulness model, such as self-acceptance, purpose in life, relation to others, and psychological flexibility. Future studies should confirm such conclusions by extending the sample to male subjects, implementing a randomized control group, and incorporating the putative mediator variables in the treated and control groups.

## Figures and Tables

**Figure 1 ijerph-18-05512-f001:**
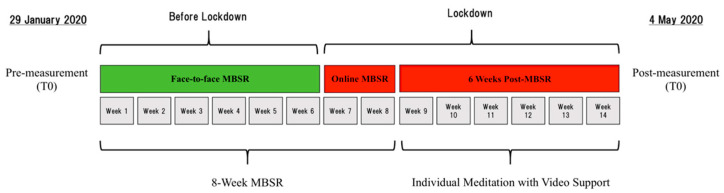
Timeline of the mindfulness-based training that was delivered to a female sample during the COVID-19 health emergency. At T0, the MBSR training started with group meetings (for six weeks) and continued with two online lessons via the internet due to the lockdown. After the end of the 8-week MBSR, participants received support for meditation practice with online videos for a further 6 weeks until the end of the lockdown (T1).

**Figure 2 ijerph-18-05512-f002:**
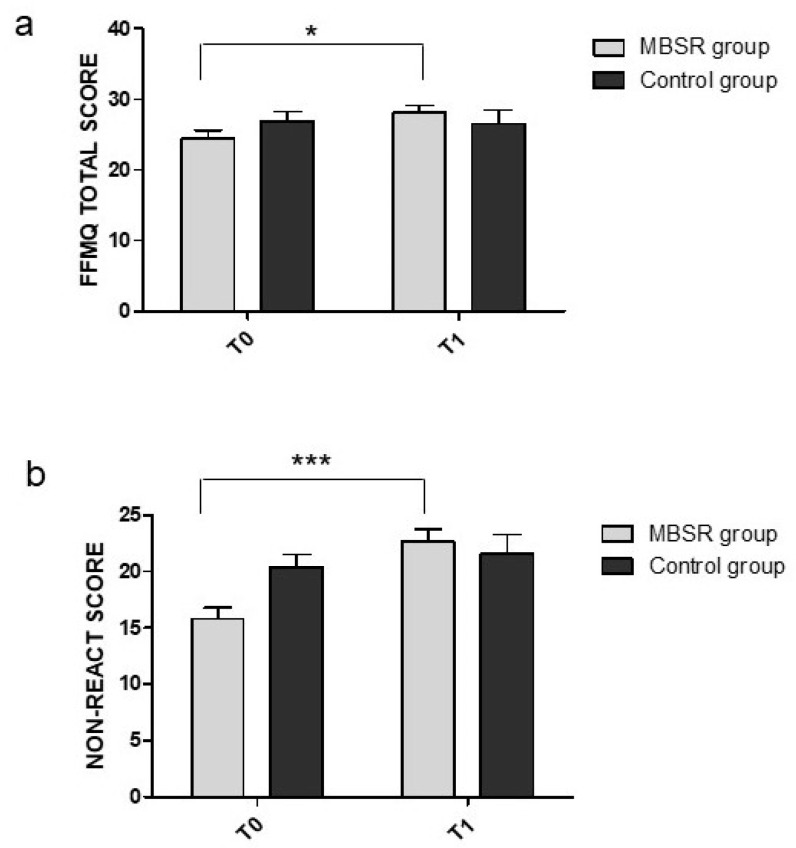
Mean changes in the (**a**) FFMQ total score and (**b**) NRE subscale score. All measures are reported as mean ± SEM. * *p* < 0.05, *** *p* < 0.001.

**Figure 3 ijerph-18-05512-f003:**
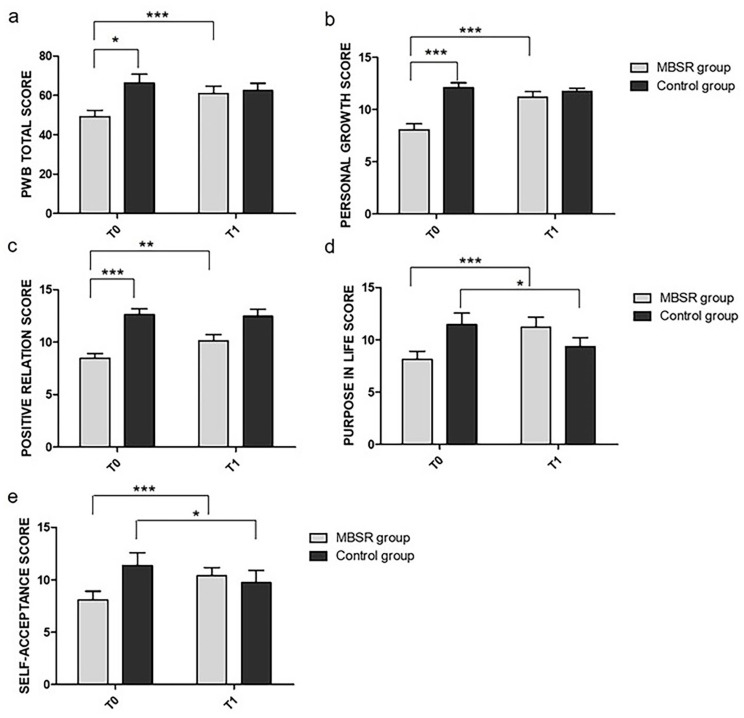
Mean change in the (**a**) PWB total score, (**b**) PG subscale score, (**c**) PR subscale score, (**d**) PL subscale score, and (**e**) SA subscale score. All measures are reported as mean ± SEM. * *p* < 0.05, ** *p* < 0.01, *** *p* < 0.001.

**Figure 4 ijerph-18-05512-f004:**
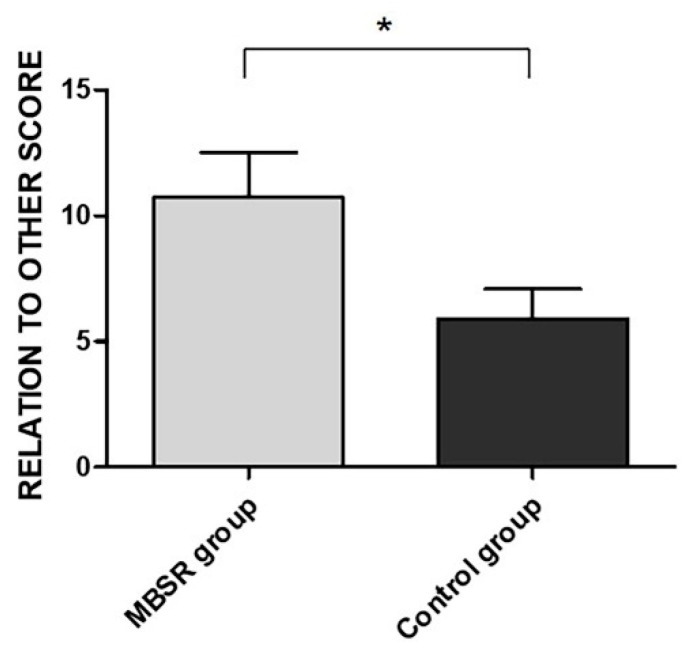
Mean difference between the MBSR and control groups in the RO subscale. All measures are reported as mean ± SEM. * *p* < 0.05.

**Figure 5 ijerph-18-05512-f005:**
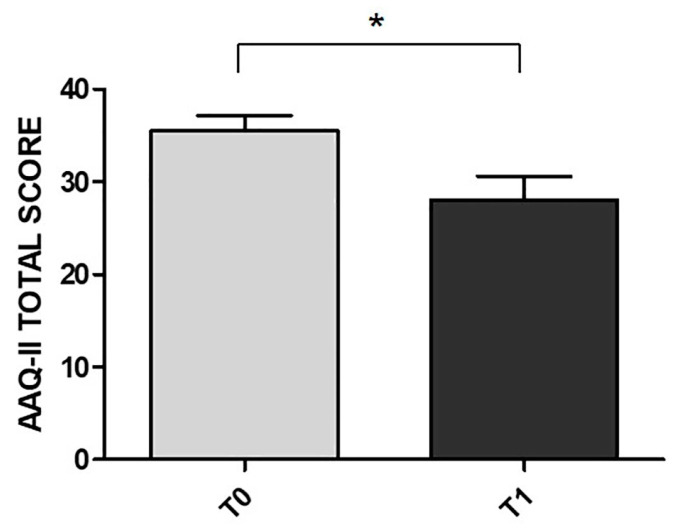
Mean difference between T0 and T1 in the AAQ-II total score of the MBSR group. All measures are reported as mean ± SEM. * *p* < 0.05.

**Figure 6 ijerph-18-05512-f006:**
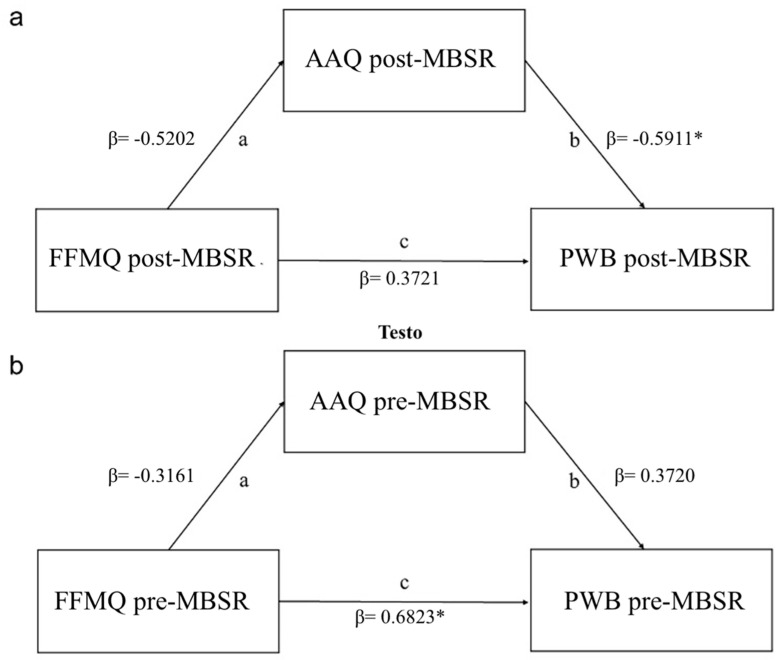
Path analysis illustrating (**a**) the mediation effect of flexibility at T1 on the relationship between mindfulness skills (FFMQ post-MBSR) and psychological well-being (PWB post-MBSR) and (**b**) the absence of a mediation effect of flexibility at T0 on the relationship between mindfulness skills (FFMQ pre-MBSR) and psychological well-being (PWB pre-MBSR). * *p* < 0.05.

## Data Availability

The datasets used and/or analyzed during the current study are available from the corresponding author on reasonable request.

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
