# Peer review of "Beneficial Effects of Mindfulness-Based Stress Reduction Training on the Well-Being of a Female Sample during the First Total Lockdown Due to COVID-19 Pandemic in Italy"

_ijerph, 2021, doi:10.3390/ijerph18115512_

Round 1

Reviewer 1 Report

I can not accept the answer for my remark: "unfortunately, it is impossible to replicate the unique and particular conditions in which we conducted the study". Pandemic situation lasted so long and used methods (self - report, Internet gained data) were so easy to applicate, that there were enough opportunity to gather more data.   

Author Response

Reviewer 1

I can not accept the answer for my remark: "unfortunately, it is impossible to replicate the unique and particular conditions in which we conducted the study". Pandemic situation lasted so long and used methods (self - report, Internet gained data) were so easy to applicate, that there were enough opportunity to gather more data.  

Answer: We must reiterate the impossibility of replicating the study as is. Indeed, it boasts a unique and unrepeatable condition: data collection before the first Italian lockdown, when the Sars-Cov2 virus was not yet widespread throughout the world, except in China. How would it be possible to replicate this condition? While we acknowledge that the current pandemic is long-lasting, our results refer specifically to the first Italian lockdown, i.e., the period in which every daily activity was suspended, with a significant effect on psychological well-being. How would it be meaningful to integrate new data collected in a diverse period, characterized by a partial return to normality, when the psychological state may differ?

Reviewer 2 Report

Few sentences should be added to inform the reader about the mediation analysis method by Barron and Kenney, and how this analysis is performed and which the underlying assumptions.

Answer: It is evident from the literature that mindfulness is associated with enhanced flexibility, and greater flexibility is linked with better health and well-being. This observation lends indirect support to our speculation (that led us to do the mediational analysis) that the beneficial health effect of mindfulness may be mediated by flexibility. The mediation analysis was conducted by running a hierarchical regression analysis using the SPSS script developed by Hayes (2009). For this analysis, the total score on the FFMQ questionnaire was considered as a predictor and the global psychological well-being index of the PWB questionnaire was taken as the criterion variable. The mediated relationship of mindfulness and mental health via flexibility was then calculated. A mediator mediates the relationship between the independent and dependent variables explaining the reason for such a relationship to exist This information was incorporated into the text (255-260).

Comment from the reviewer

A correct mediation analysis must incorporate the measurement of the mediator both in the treated and in the control group.

If the authors are able to make reasonable assumptions and or have additional information they should explain using causal inference terminology how these assumptions are able to amend the lack of measurement of the mediator also in controls.

Please look at the book on causal inference

Causality: Statistical Perspectives and Applications | Wiley

The justification given in the paper does not hold.

The same kind of reasoning applies also to the absence of randomization.

In the absence of randomization and of a propensity score analysis we can never be sure that the comparisons of the two groups is not biased. A posteriori reasoning on the possible characteristics of a non randomized controlled group does not help to avoid the possibility of a bias.

In absence of a randomized control group and in the absence of a propensity score analysis it is not possible to draw any causal inference on the effect of the treatment.

These limitations must be clearly explained and acknowledged by the authors of the paper in the limitation paragraph.

Again look for reference to a book on causality :

Causality: Statistical Perspectives and Applications | Wiley

Author Response

Reviewer 2

A correct mediation analysis must incorporate the measurement of the mediator both in the treated and in the control group. If the authors are able to make reasonable assumptions and or have additional information they should explain using causal inference terminology how these assumptions are able to amend the lack of measurement of the mediator also in controls. Please look at the book on causal inference. Causality: Statistical Perspectives and Applications | Wiley. The justification given in the paper does not hold. The same kind of reasoning applies also to the absence of randomization. In the absence of randomization and of a propensity score analysis we can never be sure that the comparisons of the two groups is not biased. A posteriori reasoning on the possible characteristics of a non randomized controlled group does not help to avoid the possibility of a bias. In absence of a randomized control group and in the absence of a propensity score analysis it is not possible to draw any causal inference on the effect of the treatment. These limitations must be clearly explained and acknowledged by the authors of the paper in the limitation paragraph. Again look for reference to a book on causality : Causality: Statistical Perspectives and Applications | Wiley

Answer: In the latest version of the Limitations section, thanks to the valuable comments of the reviewer, we have discussed more thoroughly the potential issues arising from the lack of randomization and the absence of a control group in the mediation analysis. We have also added additional limitations we felt appropriate to mention (lines 512-513; 517-519; 522-530; 543-549). Furthermore, we toned down the causality claims in the “Abstract” (lines 25-27), in the paragraph “Objective 4: The role of psychological flexibility as mediator variable” (lines 493-498; 507) and in the “Conclusions” (lines 550-558).

Reviewer 3 Report

The authors responded thoroughly to reviews, although the study's overall contribution still seems to me rather modest, given the significant limitations discussed in the reviews and responses.

Author Response

Reviewer 3

The authors responded thoroughly to reviews, although the study's overall contribution still seems to me rather modest, given the significant limitations discussed in the reviews and responses.

We thank the reviewer for recognizing our effort to address the paper’s limitations. However modest, we believe that our study deserves publication because it suggests MBSR can support psychological well-being, at least in female subjects, even during an unrepeatable adverse condition such as a pandemic lockdown. Moreover, while inconclusive, our results may stimulate further research on psychological processes possibly mediating such support.

Round 2

Reviewer 1 Report

I accept your explanation, but I expect in such case, that in the limitation part of your work, it will be underlined as a serious weakness, which should be taken into consideration in analyzing results. 

Reviewer 2 Report

I'm happy about the revision made to the work

Now the paper can be pubblished without any more revisions